# Relationship between working conditions and psychological distress experienced by junior doctors in the UK during the COVID-19 pandemic: a cross-sectional survey study

Alice Dunning ![ORCID],[1] Kevin Teoh ![ORCID],[2] James Martin,[3] Johanna Spiers ![ORCID],[4] Marta Buszewicz,[5] Carolyn Chew-Graham ![ORCID],[6] Anna Kathryn Taylor ![ORCID],[7] Anya Gopfert,[8] Maria Van Hove,[9] Louis Appleby,[10] Ruth Riley ![ORCID] [11]

For numbered affiliations see end of article.

**Correspondence to**
Alice Dunning;
alice.dunning@bthft.nhs.uk

## ABSTRACT

**Objectives** This paper explored the self-reported prevalence of depression, anxiety and stress among junior doctors during the COVID-19 pandemic. It also reports the association between working conditions and psychological distress experienced by junior doctors.

**Design** A cross-sectional online survey study was conducted, using the 21-item Depression, Anxiety and Stress Scale and Health and Safety Executive scale to measure psychological well-being and working cultures of junior doctors.

**Setting** The National Health Service in the UK.

**Participants** A sample of 456 UK junior doctors was recruited online during the COVID-19 pandemic from March 2020 to January 2021.

**Results** Junior doctors reported poor mental health, with over 40% scoring extremely severely depressed (45.2%), anxious (63.2%) and stressed (40.2%). Both gender and ethnicity were found to have a significant influence on levels of anxiety. Hierarchical multiple linear regression analysis outlined the specific working conditions which significantly predicted depression (increased demands ($\beta$=0.101), relationships ($\beta$=0.27), unsupportive manager ($\beta$=−0.111)), anxiety (relationships ($\beta$=0.31), change ($\beta$=0.18), demands ($\beta$=0.179)) and stress (relationships ($\beta$=0.18), demands ($\beta$=0.28), role ($\beta$=0.11)).

**Conclusions** The findings illustrate the importance of working conditions for junior doctors' mental health, as they were significant predictors for depression, anxiety and stress. Therefore, if the mental health of junior doctors is to be improved, it is important that changes or interventions specifically target the working environment rather than factors within the individual clinician.

## STRENGTHS AND LIMITATIONS OF THIS STUDY

⇒ The survey was delivered between March 2020 and January 2021, and therefore data were included, both peak and non-peak, in infection rates of the pandemic, meaning the results are significant to the study aims.

⇒ The study design links participants' psychological distress against an established and validated measure of working cultures.

⇒ The sample of participants showed diverse representation across ethnicity, specialties and years of experience.

⇒ However, the sample was not proportionately representative, as the majority of participants were white and female.

## BACKGROUND

The demands of workload and job expectations on healthcare workers have been widely acknowledged.[1] These demands have been further exacerbated by the COVID-19 pandemic, which created unprecedented working conditions for these workers.[2] Prior to the pandemic, the 2019 National Health Service (NHS) staff survey found that 40% staff reported feeling unwell due to work-related stress in 5 years.[3] The 2019 NHS Long Term Plan[4] outlined the impact of increased demands on healthcare staff well-being. Ten years previously, the NHS Health and Well-Being report argued that staff well-being should be at the centre of all organisations in the NHS.[5] However, despite acknowledgement of the demanding work environments and poor well-being among healthcare workers in the NHS, relatively little has been done to better support staff.[6]

Doctors report more anxiety and depression than the general population.[7] A particularly vulnerable group of doctors experiencing distress are junior doctors. There are currently 65 542 junior doctors working in the NHS,[8] this includes qualified doctors in training or doctors who are not consultants; this title also includes foundation year doctors.[9] The General Medical Council has highlighted the

low morale, distress and alienation experienced by junior doctors.[10] Junior doctors and medical students reported the highest rates of being diagnosed with a mental illness, including anxiety and depression during the previous 12 months compared with others doctors; of these, 91% of junior doctors were at high or very high risk of burnout.[11] Furthermore, Irish junior doctors reported significantly higher levels of psychological distress in comparison with consultants.[12] Junior doctors have also reported feeling dehumanised by their employers.[13] Evidence suggests that junior doctors are more vulnerable to feeling distressed which are attributed to feelings of being unsupported, devalued and without having autonomy.[14] A barrier for junior doctors seeking help for mental health concerns is the fear of failing if off sick.[15]

Research has identified gender differences in the mental health of UK doctors; in 2019 the gender split of all UK doctors was 45% female and 55% male.[16] One survey found that women showed a higher risk of burnout and were 50% more likely to report a mental illness diagnosis compared with their male counterparts.[11] Although there are mixed findings about suicide rates internationally, there are concerns over suicidality among doctors, particularly among female doctors.[17]

Frontline healthcare professionals have reported increased levels of psychological distress while working during the COVID-19 pandemic.[18–23] Furthermore, research has demonstrated the increased risk of burnout for NHS doctors at this time, with nearly half describing deterioration of their mental health.[24] High levels of psychological distress have been reported among anaesthesiology, emergency medicine and intensive care doctors in both Ireland and the UK.[25] The increased work demands, redeployment, inadequate access to basic facilities or safe spaces to rest, loss of autonomy and anxiety around the plans for recovery were identified as contributing factors to the deterioration of their mental health.[24 26] The pandemic has also had a significant impact on training and further exacerbated the levels of poor mental health among junior doctors, with 70% experiencing anxiety, depression and burnout.[27]

### Working conditions as a predictor of junior doctors' mental health

In light of the prevalence of poor mental health among junior doctors, it is important to understand the relevant antecedent factors. While workplace health interventions typically focus on addressing individual-level factors in the behaviour or approach of junior doctors (eg, coping behaviours and resilience), there is increasing recognition that how workplaces are organised, designed and managed is a more important contributing factor, not only to doctors' mental health but also towards patient care.[28 29] Two work-related factors, including high demands and a poor environment, were identified as being associated with higher levels of burnout among junior doctors within a systematic review of 48 studies.[30] An Australian survey found that increasing working hours led to an increase in mental health problems among junior doctors.[7] Qualitative research in Australia identified support from colleagues as improving junior doctors' levels of burnout and well-being.[31] Similarly, interviews with junior doctors in the UK identified poor working conditions, toxic work cultures, lack of support and mental health stigma as being contributing factors to poor levels of well-being among junior doctors.[32]

To better, and more holistically, support the mental health of junior doctors, it is important to understand the specific working conditions within the UK health system related to their mental health. Although other studies have correlated other measures of work environment with the mental health of NHS doctors (eg, 28) or nurses,[33 34] this has not been done specifically for junior doctors.

### Aims

To explore the prevalence of depression, anxiety and stress experienced by junior doctors in the UK during the COVID-19 pandemic, and to identify working conditions associated with psychological distress.

## METHOD
### Study design and sample

A cross-sectional online survey, conducted as part of a wider mixed-methods study,[31 35] was used to measure the working conditions and mental health of a convenience sample of UK junior doctors. Participants were recruited online from March 2020 until January 2021 through advertisements on social media posts, junior doctor forums and emails circulated via specialist schools. The survey was hosted on the online platform Qualtrics, which allowed participants to access the questionnaire at a time which suited them. Participants were entered into a prize draw for shopping vouchers (three prizes of £50 were available) to thank them for their time.

## MEASURES
### Demographic information

Age, gender, ethnicity, sexuality, years as a doctor and specialty were recorded at the beginning of the online survey.

### Mental health

Levels of depression, anxiety and stress were all measured using the 21-item Depression, Anxiety and Stress Scale.[36] Participants rated the extent to which they had experienced each item within the past week on a 4-point Likert scale. Following the scale guidelines, participant scores for depression, anxiety and stress were multiplied by two, and categorised into normal, mild, moderate, severe and extremely severe, following the recommended cut-off scores.

### Working conditions

The UK Health and Safety Executive's management standards framework was used. This identifies seven key

aspects of the working environment using a 35-item scale: role clarity, peer support, strained relationships, managerial support, work demands, control over working environment and change. These are based on relevant occupational health psychology theories (eg, the job demands resources model).[37] While the framework has been widely applied across different sectors (eg, 38–40), we are not aware of its application in the NHS. Participants were asked to rate how often they experienced each item within the last 6 months on a 5-point Likert scale.

### Patient and public involvement
No patient involvement.

### Data analysis
After data were extracted from Qualtrics, we carried out data management and analysis using SPSS V.26. Measures were scored according to their instructions and, where appropriate, items reverse scored so that a higher score represents a stronger measure of the construct. Participant demographics were summarised using numbers and percentages, means and SDs or median and IQRs, as appropriate. A Shapiro-Wilk test was used to assess normality. The normality of the variables was not established as all the measures were skewed and had a kurtosis score greater than the absolute value of 2.0. Little's $X^2$ statistic for testing whether values are missing completely at random[41] was not significant ($X^2$ (1746, 456)=1145.09, p=1.00). This demonstrates that there was no pattern to the missing values within the data set. To test for association between participants' characteristics and the depression, anxiety, and stress scores, we used Kruskall-Wallis non-parametric tests with Bonferroni adjustments to test against the raw scores, and $X^2$ to test against the classified scores. We then carried out Spearman's r correlations between the study variables as assumptions of normality were not met.

To quantify the effect of working conditions on each of the outcomes (depression, anxiety and stress), we fitted two linear regression models to each outcome. Our first model included only the participants' demographics (age, gender, years as a doctor). The second model included participants' demographics and the seven working conditions (demands, control, manager support, peer support, role, change, strained relationships).

## RESULTS
### Participant characteristics and levels of depression, anxiety and stress
A total of 456 responses were received with the mean age of 30.70 years (SD=4.87). The mean number of years working as a doctor was 4.94 (SD=3.56). Most participants identified as female (67.1%) and came from a white background (71.1%), 80% of the sample identified as heterosexual. A full demographic breakdown is presented in table 1.

Table 1 provides an overview of the proportion of the sample that was categorised on each degree of the mental health measures. Across all three measures, at least 70% of the sample scored 'severe' or 'extremely severe'. For depression, 45.2% of participants were classed as having extremely severe levels of depression, with an additional 23.3% classed as severe (p<0.001). There were more participants classed as having extremely severe levels of anxiety (63.2%) compared with those with severe (25.2%) or moderate (11.7%) levels (p<0.001). No participants had normal or mild levels of depression or anxiety. More participants also reported extremely severe (40.2%) or severe (31.5%) levels of stress than less severe levels (p<0.001). $X^2$ analysis between participant characteristics (ie, gender, ethnicity, sexuality) and classification of depression, anxiety and stress showed no significant associations (table 1).

We found no evidence of differences on levels of depression (p=0.439) and stress (p=0.290) between male and female participants (table 2). However, significant differences were reported for anxiety (p=0.032), where participants who identified as female (M=25.45) reported a higher mean score for anxiety than male (M=23.71). No differences were observed based on their sexual orientation for levels of depression (p=0.098), anxiety (p=0.260) or stress (p=0.662).

In terms of ethnicities, a Kruskall-Wallis (table 3) found a significant difference in levels of depression (p=0.016). Post-hoc comparison using Bonferroni test showed participants from mixed (M=24.80) background reporting lower levels of depression than those from an Asian (M=31.51) or white (M=27.74) background. Participants from a white (M=27.74) background also reported lower levels of depression than those from other (M=34.76) and from an Asian (M=31.55) background. Differences were also observed across ethnicities for anxiety (p=0.020), with Bonferroni comparisons showing that participants from an Asian background (M=27.88) reported higher levels of anxiety than those from a white background (M=24.16). No differences were observed for stress (p=0.770) between participants from different ethnic groups.

### Descriptive statistics and relationships for perceived working conditions, depression, anxiety and stress
Table 2 presents the descriptive statistics and internal reliabilities for all variables in this study, and that all composite measures had adequate internal consistency (α>0.70). Bivariate two-tailed non-parametric correlations show the presence of all anticipated relationships in the appropriate directions. More specifically, reporting more challenging working conditions (ie, high demands, strained relationships) was associated with lower levels of positive working conditions (ie, control, manager support, peer support, role, change), as well as higher levels of depression, anxiety and stress. Higher levels of the four positive working conditions were correlated with lower levels of junior doctors' levels of depression, anxiety and stress.

**Table 1** Frequency and percentages of participant demographics mapped against degree of depression, anxiety and stress

| | N | Depression | | | | | Anxiety | | | | | Stress | | | | | |
|---|---|---|---|---|---|---|---|---|---|---|---|---|---|---|---|---|---|
| | | Normal/mild | Moderate | Severe | Extremely severe | P value | Normal/mild | Moderate | Severe | Extremely severe | P value | Normal | Mild | Moderate | Severe | Extremely severe | P value |
| Total | 456 (100%) | 0 (0%) | 135 (31.4%) | 100 (23.3%) | 194 (45.2%) | 0.01* | 0 (0%) | 50 (11.7%) | 108 (25.2%) | 271 (63.2%) | 0.01* | 8 (1.9%) | 39 (9.1%) | 74 (17.3%) | 135 (31.5%) | 172 (40.2%) | 0.01* |
| Gender | | | | | | 0.78 | | | | | 0.22 | | | | | | 0.67 |
| Male | 123 (27%) | – | 33 (28.7%) | 30 (26.1%) | 52 (45.2%) | | – | 16 (13.9%) | 35 (30.4%) | 64 (55.7%) | | 3 (2.6%) | 11 (9.6%) | 25 (21.7%) | 33 (28.7%) | 43 (37.4%) | |
| Female | 306 (67.1%) | – | 96 (33.1%) | 65 (22.4%) | 129 (44.5%) | | – | 31 (10.7%) | 72 (24.8%) | 187 (64.5%) | | 4 (1.4%) | 28 (9.7%) | 45 (15.6%) | 97 (33.6%) | 115 (39.8%) | |
| Prefer to self-describe | 4 (0.9%) | – | 1 (33.9%) | 1 (33.9%) | 1 (33.9%) | | – | – | – | 3 (100%) | | – | – | 1 (33.9%) | 1 (33.9%) | 1 (33.9%) | |
| Prefer not to say | 23 (5%) | – | 1 (14.3%) | 1 (14.3%) | 5 (71.4%) | | – | 2 (28.6%) | – | 5 (71.4%) | | – | – | 2 (28.6%) | – | 5 (71.4%) | |
| Ethnicity | | | | | | 0.28 | | | | | 0.07 | | | | | | 0.93 |
| White | 324 (71.1%) | – | 100 (32.8%) | 72 (23.6%) | 133 (43.6%) | | – | 39 (12.8%) | 83 (27.2%) | 183 (60%) | | 7 (2.3%) | 27 (8.9%) | 53 (17.4%) | 99 (32.6%) | 118 (38.8%) | |
| Mixed | 26 (6/4%) | – | 10 (38.5%) | 8 (30.8%) | 8 (30.8%) | | – | 2 (7.7%) | 9 (34.6%) | 15 (57.7%) | | – | 3 (11.5%) | 3 (11.5%) | 10 (38.5%) | 10 (38.5%) | |
| Asian | 72 (15.8%) | – | 18 (26.1%) | 16 (23.2%) | 35 (50.7%) | | – | 4 (5.8%) | 11 (15.9%) | 54 (78.3%) | | 1 (12.5%) | 7 (10.1%) | 10 (14.5%) | 21 (30.4%) | 30 (43.5%) | |
| Black | 9 (2%) | – | 3 (37.5%) | 2 (25%) | 3 (37.5%) | | – | – | 2 (25%) | 6 (75%) | | – | 1 (12.5%) | 2 (25%) | 2 (25%) | 3 (37.5%) | |
| Other | 25 (5.5%) | – | 4 (19%) | 2 (9.5%) | 15 (71.4%) | | – | 5 (23.8%) | 3 (14.3%) | 13 (61.9%) | | – | 1 (4.8%) | 6 (28.6%) | 3 (14.3%) | 11 (52.4%) | |
| Sexuality | | | | | | 0.42 | | | | | 0.49 | | | | | | 0.17 |
| Heterosexual | 365 (80%) | – | 113 (32.8%) | 79 (23%) | 152 (44.2%) | | – | 43 (12.5%) | 89 (25.9%) | 212 (61.6%) | | 8 (2.3%) | 33 (84.6%) | 57 (31.2%) | 107 (31.2%) | 138 (40.2%) | |
| Lesbian, gay, bisexual, and transgender (LGBTQ+) | 69 (15.2%) | – | 18 (27.7%) | 18 (27.7%) | 29 (44.6%) | | – | 4 (6.2%) | 17 (26.2%) | 44 (67.7%) | | – | 5 (7.7%) | 10 (15.4%) | 26 (40%) | 24 (36.9%) | |
| Prefer not to say | 21 (4.6%) | – | 3 (20%) | 2 (13.3%) | 10 (66.7%) | | – | 2 (13.3%) | 2 (13.3%) | 11 (73.3%) | | – | 1 (6.7%) | 6 (40%) | 1 (6.7%) | 7 (46.7%) | |

*means that it is a significant i.e. <0.05.

**Table 2** Descriptive statistics, correlations and reliability coefficients among study variables

| Variable | Internal consistency (Cronbach's α) | 1 | 2 | 3 | 4 | 5 | 6 | 7 | 8 | 9 | 10 | 11 | 12 |
|---|---|---|---|---|---|---|---|---|---|---|---|---|---|
| 1. Years as doctor | – | 4.94 (3.56) | 0.70** | –0.09* | 0.15** | 0.10* | 0.04 | 0.04 | 0.20** | 0.09 | –0.02 | –0.17** | –0.03 |
| 2. Age | – | | 30.70 (4.87) | –0.04 | 0.10* | 0.04 | –0.06 | 0.07 | 0.06 | –0.01 | 0.03 | –0.16** | –0.04 |
| 3. Demands | 0.86 | | | 3.19 (0.68) | –0.60** | –0.52** | –0.45** | 0.53** | –0.42** | –0.43** | 0.40** | 0.34** | 0.45** |
| 4. Control | 0.84 | | | | 2.65 (0.71) | 0.66** | 0.52** | –0.42** | 0.43** | 0.58** | –0.37** | –0.29** | –0.34** |
| 5. Manager support | 0.90 | | | | | 2.98 (0.95) | 0.71** | –0.54** | 0.51** | 0.66** | –0.46** | –0.30** | –0.39** |
| 6. Peer support | 0.84 | | | | | | 3.61 (0.73) | –0.56** | 0.48** | 0.46** | –0.45** | –0.32** | –0.38** |
| 7. Strained relationships | 0.84 | | | | | | | 2.34 (0.) | –0.45** | –0.41** | 0.49** | 0.36** | 0.42** |
| 8. Role | 0.74 | | | | | | | | 3.80 (0.58) | 0.46** | –0.36** | –0.25** | –0.35** |
| 9. Change | 0.80 | | | | | | | | | 2.42 (0.81) | –0.37** | –0.20** | –0.33** |
| 10. Depression | 0.93 | | | | | | | | | | 28.50 (11.47) | 0.56** | 0.72** |
| 11. Anxiety | 0.79 | | | | | | | | | | | 25.15 (10.41) | 0.70** |
| 12. Stress | 0.88 | | | | | | | | | | | | 31.14 (9.63) |

Mean (SD) is reported diagonally.
**P<0.01 level (two tailed); *p<0.05 level (two tailed).

## Direct effects between perceived working conditions and levels of depression, anxiety and stress

Table 4 presents report that junior doctors' strained relationships (β=0.27) positively predicted levels of depression. Having good peer support was negatively associated with levels of depression (β=−0.15). There was no evidence of an association between participants' characteristics (age, gender and years as a doctor) and depression.

For anxiety, junior doctors who identified as female reported higher levels of anxiety (β=0.11) than those who identified as male. Table 4 shows that strained relationships were the strongest predictor (β=0.31) of anxiety, followed by how well change is managed (β=0.18).

Focusing on stress, we see that demographic factors made no contribution to the explained variance in levels of stress. Strained relationships (β=0.18) and demands (β=0.28) positively predicted stress. In addition, role clarity was associated with lower levels of stress (β=0.11).

## DISCUSSION

The findings show that junior doctor survey respondents reported high levels of depression, anxiety and stress during the pandemic, with at least 70% reporting 'severe' or 'extremely severe' across each measure. This study expands what has already been described in the literature by showing that specific working conditions are associated with junior doctors' levels of psychological distress, namely depression (strained relationships, managerial support), anxiety (strained relationships, organisational change, high demands) and stress (strained relationships, high demands and role).

The high levels of depression, anxiety and stress which were reported by junior doctors in this study build on previous literature which reports elevated psychological distress among junior doctors as compared with pre-pandemic levels.[12] Crucially, these figures are higher than pre-pandemic comparisons of samples including doctors in the NHS[42] and Ireland,[11] reflecting the increased strain which healthcare staff experienced while working on the front line.[2 24 27]

Furthermore, female junior doctors in this study reported higher levels of anxiety compared with their male counterparts. As increased levels of anxiety are associated with suicidality,[43] this link may provide some insight into the elevated risk of suicide among female doctors.[12] This difference in anxiety between women and men may be explained by work-related factors such as poorer work–life balance.[13] Further research should explore levels of suicidality among junior doctors as well as potential gender differences. There were also significant differences between levels of anxiety experienced by different ethnicities within this study, with junior doctors from an Asian background reporting higher levels than junior doctors from a white background. While no significant differences were observed for other ethnic groups, ethnic minorities in almost all instances reported poorer levels of mental health. It may be that the lack of

**Table 3** Mean comparisons between participants' characteristics for levels of depression, anxiety and stress

| Characteristic | Depression | | | Anxiety | | | Stress | | |
| --- | --- | --- | --- | --- | --- | --- | --- | --- | --- |
| | Mean | SD | P value | Mean | SD | P value | Mean | SD | P value |
| Gender | 28.21 | 11.30 | 0.44 | 24.94 | 10.32 | 0.03 | 30.97 | 9.60 | 0.29 |
| Male | 28.91 | 11.47 | | 23.71 | 10.58 | | 30.37 | 10.25 | |
| Female | 27.92 | 11.23 | | 25.45 | 10.18 | | 31.21 | 9.32 | |
| Ethnicity | 28.50 | 11.47 | 0.01 | 25.15 | 10.41 | 0.01 | 31.14 | 9.63 | 0.77 |
| White | 27.74 | 27.80 | | 24.16 | 9.77 | | 30.81 | 9.45 | |
| Asian | 31.51 | 12.84 | | 27.88 | 11.42 | | 31.82 | 10.08 | |
| Black | 26.75 | 10.79 | | 32.50 | 13.17 | | 31.00 | 11.76 | |
| Mixed | 24.80 | 9.48 | | 25.04 | 20.23 | | 30.72 | 8.18 | |
| Other | 34.76 | 12.70 | | 28.00 | 12.79 | | 34.19 | 11.52 | |
| Sexuality | 28.51 | 11.52 | 0.10 | 25.12 | 10.43 | 0.26 | 31.10 | 9.64 | 0.66 |
| Heterosexual | 28.17 | 11.37 | | 24.76 | 10.24 | | 30.90 | 9.73 | |
| Lesbian, gay, bisexual, and transgender (LGBTQ+) | 28.83 | 11.88 | | 26.21 | 10.93 | | 32.12 | 9.15 | |
| Prefer not to say | 34.93 | 12.23 | | 28.67 | 12.32 | | 31.20 | 9.97 | |

P values are for differences between means within each characteristic.

statistical power led to the lack of more significant differences being found. This finding is supported by previous literature, which outlines the increased risk of mental health problems for ethnic minorities.[44] This study is the first to demonstrate an elevated risk of poor mental health among junior doctors with an Asian background. This difference between ethnicities may be explained by poorer working conditions which are experienced by ethnic minorities,[45] or the increased likelihood of experiencing harassment in the workplace.[32 46] Future research should explore whether there are additional factors associated with differences between gender, ethnicity and anxiety; such data could support targeted changes or interventions to support any at-risk group.

### Working conditions as an antecedent to junior doctors' mental ill-health

The most consistent predictor of depression, anxiety and stress was strained relationships. Previous literature supports this finding, as bullying or witnessing harassment has been found to impact NHS healthcare staff well-being and job satisfaction.[47] Furthermore, toxic working cultures, including bullying, discrimination and a blame culture, were found to be the most frequently discussed source of distress for junior doctors.[32] A lack of supportive colleagues during incidents or through discontinuity in teams was also highlighted by Riley and colleagues,[35] as a contributing factor to psychological distress in junior doctors.

Additionally, an online survey found the most reported factor for boosting junior doctors' morale in the UK was feeling part of a team,[48] demonstrating the impact positive relationships can have. Further, belonging, and feeling valued and connected with colleagues and supervisors/consultants/mangers are protective and are associated

with better morale and positive mental health among staff.[47] In line with these findings, interventions which centre on creating supportive working relationships, such as peer mentoring programmes, have been found to have a positive effect on psychological well-being and job satisfaction for junior doctors.[48]

Belonging, feeling valued and connectedness are protective factors and can promote positive mental health and a positive professional identity; for instance, peer/colleague support in the workplace has been found to mitigate the negative psychological consequences of adverse events. In line with the evidence that supportive relationships with colleagues are important for mitigating psychological distress, supportive managers were found to be significantly associated with lower levels of depression in this study. The importance of supportive leadership for healthcare staff[49] and doctors'[29] well-being has been previously identified. Supportive relationships and feeling valued by supervisors/consultants act as a buffer against the demands of the job.[35] Taken together, the findings that strained relationships and supportive leadership are associated with mental health demonstrate the need for a supportive environment, in which junior doctors work within consistent teams under good leadership to be adopted.

Intense work demand was also associated with psychological distress, demonstrating its importance for junior doctors' well-being. Previous literature has outlined the increasing strain on healthcare workers because of the demands of their role,[1] which were exacerbated through the pandemic.[2 24] Considering the demands healthcare staff face, the prevalence of poor mental health found in this study is unsurprising. This association between high/intense job demands and poor mental health is

**Table 4** Multiple linear regression analyses with depression, anxiety and stress as dependent variables

| Predictors | Step 1 | | Step 2 | |
|---|---|---|---|---|
| | β | B (95% CI) | β | B (95% CI) |
| **Depression** | | | | |
| Age | 0.048 | 0.110 (−0.216 to 0.435) | 0.007 | 0.017 (−0.259 to 0.292) |
| Gender (female) | −0.027 | 0.675 (−3.156 to 1.807) | −0.012 | −0.305 (−2.400 to 1.791) |
| Years as doctor | −0.036 | −0.110 (−0.0543 to 0.324) | 0.004 | 0.014 (−0.359 to 0.386) |
| Demands | | | 0.101 | 1.683 (−0.187 to 3.552) |
| Control | | | −0.017 | −0.273 (−2.245 to 1.699) |
| Manager support | | | −0.111 | −1.323 (−3.066 to 0.420) |
| Peer | | | −0.147 | −2.266 (-4.251 to 0.281) |
| Strained relationships | | | 0.273 | 3.954 (2.314 to 5.593) |
| Role | | | −0.088 | −1.699 (−3.775 to 0.377) |
| Change | | | 0.039 | 0.550 (−1.107 to 2.208) |
| $R^2$ | 0.002 | | 0.324 | |
| $\Delta R^2$ | 0.002 | | 0.322 | |
| P value | 0.845 | | 0.001 | |
| **Anxiety** | | | | |
| Gender (female) | 0.079 | 1.784 (−0.443 to 4.000) | 0.108 | 2.457 (0.458 to 4.457) |
| Years as doctor | −0.095 | −0.266 (0.653 to 0.122) | −0.083 | −0.233 (−0.589 to 0.122) |
| Age | −0.088 | −0.184 (−0.475 to 0.106) | −0.120 | −0.251 (−0.514 to 0.012) |
| Demands | | | 0.179 | 2.716 (0.932 to 4.500) |
| Control | | | 0.028 | 0.413 (−1.469 to 2.295) |
| Manager support | | | 0.009 | 0.100 (−1.563 to 1.763) |
| Peer | | | −0.105 | −1.476 (−3.369 to 0.418) |
| Strained relationships | | | 0.314 | 4.135 (2.571 to 5.699) |
| Role | | | −0.051 | −0.886 (−2.867 to 1.095) |
| Change | | | 0.104 | 1.328 (−0.3254 to 2.909) |
| $R^2$ | 0.037 | | 0.256 | |
| $\Delta R^2$ | 0.037 | | 0.219 | |
| P value | 0.002 | | 0.001 | |
| **Stress** | | | | |
| Gender (female) | 0.049 | 1.029 (−1.081 to 3.140) | 0.061 | 1.298 (−0.511 to 3.108) |
| Years as doctor | −0.034 | −0.09 (−0.459 to 0.279) | 0.022 | 0.057 (−0.266 to 0.379) |
| Age | 0.019 | 0.038 (−0.239 to 0.314) | −0.027 | −0.054 (−0.291 to 0.184) |
| Demands | | | 0.294 | 4.164 (2.549 to 5.479) |
| Control | | | 0.065 | 0.879 (−0.824 to 2.582) |
| Manager support | | | −0.034 | −0.346 (−1.851 to 1.160) |
| Peer | | | −0.103 | −1.352 (−3.067 to 0.363) |
| Strained relationships | | | 0.181 | 2.225 (0.807 to 3.644) |
| Role | | | −0.119 | −1.943 (−3.735 to 0.151) |
| Change | | | −0.007 | −0.089 (−1.521 to 1.343) |
| $R^2$ | 0.003 | | 0.304 | |
| $\Delta R^2$ | 0.003 | | 0.301 | |
| P value | 0.764 | | 0.001 | |

*$P < 0.05$, **$p < 0.01$.
CI: 95% unstandardised CIs.

supported by previous research where job demands predicted doctors' reported well-being[29] were associated with junior doctors' levels of burnout,[30] and contributed to junior doctors' psychological distress.[32] This research further highlights the need to address the demanding working conditions which are impacting on junior doctors' distress.

A key strength of this study is the diverse representation across ethnicities, different specialties and years of experience. However, most of the junior doctors in this study

were white women. Considering the gender and ethnicity differences reported in the results, a more proportionally representative sample would have improved the generalisability of the results. Furthermore, because of the use of convenience sampling recruitment methods, the results may be biased to junior doctors who access online platforms. We did not consider positive manifestations of well-being (eg, job satisfaction, work engagement) which have been linked to both doctors' working environment and to patient care.[50 51] This was beyond the scope of this funded study, but future researchers should explore a more rounded perspective of doctors' well-being. Finally, the cross-sectional study design means causality cannot be presumed, and the single time point measure also does not capture the full effects of the pandemic, with other studies showing a general deterioration of healthcare workers' mental health over the course of the pandemic.[26 52 53] As we did not factor in date of completion, we were not able to determine the proportion of respondents who completed the survey during one of the peak periods. Nonetheless, the findings from this study provide a valuable understanding of the relationship between working conditions and mental health for junior doctors.

## Conclusion

In summary, the findings of this study reflect the poor levels of psychological well-being many junior doctors experienced during the pandemic. Our results demonstrate the association between working conditions, specifically job demands, strained relationships, organisational change and role certainty, and the mental health of junior doctors. This indicates the need for systemic changes or interventions that focus on improving the working conditions of junior doctors rather than focusing on 'resilience' in the individual.

**Author affiliations**
[1]Research, Bradford Teaching Hospitals NHS Foundation Trust, Bradford, UK
[2]Department of Organizational Psychology, Birkbeck University of London, London, UK
[3]Institute of Applied Health Research, University of Birmingham, Birmingham, UK
[4]College of Medical and Dental Sciences, University of Birmingham, Birmingham, UK
[5]Research Department of Primary Care and Population Health, University College London, London, UK
[6]School of Medicine, Keele University, Keele, UK
[7]School of Medicine, University of Leeds, Leeds, UK
[8]School of Medicine, Oxford University Hospitals NHS Trust, Oxford, UK
[9]London School of Hygiene & Tropical Medicine, London, UK
[10]Department of Psychiatry & Behavioral Sciences, The University of Manchester Faculty of Medical and Human Sciences, Manchester, UK
[11]School of Health Sciences, University of Surrey, Guildford, UK

**Acknowledgements** The authors would like to thank all junior doctor respondents who kindly gave their time to complete the survey. We also wish to thank members of the PPIE group who provided valuable input throughout the study.

**Contributors** AD, KT, JM, JS, MB, CC-G, AKT, AG, MVH, LA and RR contributed to the study conception, design, data collection and approval of the final article. AD, KT, JM and RR conducted the data analysis and drafting of the article. AD submitted the final version of this article for publication and is acting as guarantor .

**Funding** The study was funded by NIHR Research for Patient Benefit (PB-PG-0418-20023).

**Disclaimer** The views and opinions expressed therein are those of the authors and do not necessarily reflect those of the NIHR, NHS or the Department of Health.

**Competing interests** None declared.

**Patient and public involvement** Patients and/or the public were not involved in the design, or conduct, or reporting, or dissemination plans of this research.

**Patient consent for publication** Not required.

**Ethics approval** Participants were given an information sheet and asked to provide online consent prior to responding to the survey. All identifiable information was stored securely and separately to non-identifiable survey data. The research was approved by the University of Birmingham ethics committee and Health Research Authority (reference number: 19/HRA/6579).

**Provenance and peer review** Not commissioned; externally peer reviewed.

**Data availability statement** No data are available. This study has not received ethical approval to share confidential data with any third party other than the study research team.

**ORCID iDs**
Alice Dunning http://orcid.org/0000-0001-5078-7567
Kevin Teoh http://orcid.org/0000-0002-6490-8208
Johanna Spiers http://orcid.org/0000-0002-3935-1997
Carolyn Chew-Graham http://orcid.org/0000-0002-9722-9981
Anna Kathryn Taylor http://orcid.org/0000-0002-8149-3841
Ruth Riley http://orcid.org/0000-0001-8774-5344

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
