## [Reviewer comments · BMJ Open]

ARTICLE DETAILS

TITLE (PROVISIONAL)	The relationship between working conditions and psychological distress experienced by junior doctors in the UK during the COVID-19 pandemic: A cross-sectional survey study
AUTHORS	Dunning, Alice; Teoh, Kevin; Martin, James; Spiers, Johanna; Buszewicz, Marta; Chew-Graham, Carolyn; Taylor, Anna; Gopfert, Anya; Van Hove, Maria; Appleby, Louis; Riley, Ruth

VERSION 1 – REVIEW

REVIEWER	Shacham, Maayan Ariel University, School of Social Work
REVIEW RETURNED	06-Mar-2022

GENERAL COMMENTS	Dear Authors, Thank you for this interesting manuscript. The manuscript deals with a topic that had been addressed by many study groups worldwide. Nonetheless, it is a topic of importance and should not be forgotten. I would add some references about psychological distress experienced by health care professionals early in the COVID-19 pandemic (the same time frame utilized in the current study), e.g. (1) Shacham, M., Hamama-Raz, Y., Kolerman, R., Mijiritsky, O., Ben-Ezra, M., & Mijiritsky, E. (2020). COVID-19 factors and psychological factors associated with elevated psychological distress among dentists and dental hygienists in Israel. International journal of environmental research and public health, 17(8), 2900. (2) De Kock, J. H., Latham, H. A., Leslie, S. J., Grindle, M., Munoz, S. A., Ellis, L., ... & O'Malley, C. M. (2021). A rapid review of the impact of COVID-19 on the mental health of healthcare workers: implications for supporting psychological well-being. BMC public health, 21(1), 1-18. In addition, as the authors reflect on doctors resilience factors, perhaps addressing females resilience should be incorporated to the discussion section (as most participants in the current study were females). Thank you again for this nice manuscript.
--

REVIEWER	Gómez Salgado, Juan Universidad de Huelva, Nursing
REVIEW RETURNED	12-Mar-2022

GENERAL COMMENTS	This is a good article in where little modifications are required. Good job.
---

	Your study extends from March 2020, to January 2021. The pandemic changed, the infections were different and the population got accustomed to the disease along the first year. ¿Could the time have changed the "minds" of the junior doctors and also of the population they received during the investigation? As you use a convenience sample, recruited by posting and social groups, this study could be biased in this way. Also, you talk about population of UK junior doctors, but I cannot find the total number of junior doctors in the UK, in order to see if your sample is representative. How did you know the minimum number of junior doctors required for a representative sample? You indicate that the sample was not proportionately representative between women and men. It is possible to indicate the total number of junior doctors women and men in the UK at the Introduction section? At the actual moment, there are many studies on the fear, anxiety and psychological distress caused by Covid 19 in healthcare professionals. The most common scales have been Golberg's GHQ12, UWES9 for Work Engagement, or the AMICO scale for fear and anxiety to Covid-19. Did you consider to use some of these scales in your study? You could add some investigations which have been based on the same variables, in order to complete your vision of the healthcare professionals during the pandemic: https://pubmed.ncbi.nlm.nih.gov/34930733/ https://www.ncbi.nlm.nih.gov/pmc/articles/PMC8250215/ https://pubmed.ncbi.nlm.nih.gov/33804351/ https://pubmed.ncbi.nlm.nih.gov/33575245/ https://pubmed.ncbi.nlm.nih.gov/33400325/ https://pubmed.ncbi.nlm.nih.gov/32699204/
--	---

VERSION 1 – AUTHOR RESPONSE

Reviewer comments	Response
Reviewer 1: Dr. Maayan Shacham, Ariel University	
The manuscript deals with a topic that had been addressed by many study groups worldwide. Nonetheless, it is a topic of importance and should not be forgotten. I would add some references about psychological distress experienced by health care professionals early in the COVID-19 pandemic (the same time frame utilized in the current study), e.g. (1) Shacham, M., Hamama-Raz, Y., Kolerman, R., Mijiritsky, O., Ben-Ezra, M., & Mijiritsky, E. (2020). COVID-19 factors and psychological factors associated with elevated psychological distress among dentists and dental hygienists in Israel. International journal of environmental research and public health, 17(8), 2900. (2) De Kock, J. H., Latham, H. A., Leslie, S. J., Grindle, M., Munoz, S. A., Ellis, L., ... & O'Malley, C. M. (2021). A rapid review of the impact of COVID-19 on the mental health of healthcare workers: implications for supporting psychological well-being. BMC public health, 21(1), 1-18.	Thank you for this lovely feedback, we have included the two suggested

	references in the introduction .
In addition, as the authors reflect on doctors resilience factors, perhaps addressing females resilience should be incorporated to the discussion section (as most participants in the current study were females).	Thank you for this comment. However, it is not clear to us where this refers to as we only mention resilience once in the discussion. In the paragraph before that (4th paragraph of the Discussion) we do provide some reflection on gender differences

	“female junior doctors in this study reported higher levels of anxiety compared with their male counterparts. As increased levels of anxiety are associated with suicidality,[43] this link may provide some insight into the elevated risk of suicide among female
--	--

	e docto rs,[12]. This differ ence in anxiet y betwe en femal es and males may be explai ned by work- relate d factor s such as poore r work life balan ce,[13]. Furth er resea rch shoul d explor e levels of suicia lity amon g junior docto rs as well as
--	--

	potential gender differences." In addition, we do mention the issue of overrepresentation of white females in the limitation section, and have now also included the gender split of doctors in the UK in the Introduction.
Reviewer: 2: Dr. Juan Gómez Salgado, Universidad de Huelva	
Your study extends from March 2020, to January 2021. The pandemic changed, the infections were different and the population got accustomed to the disease along the first year. ¿Could the time have changed the "minds" of the junior doctors and also of the population they received during the investigation?	We recognise that

	levels of mental wellbeing and perceptions have changed over the course of the pandemic; we have therefore made this point more explicit by including this as a limitation.
As you use a convenience sample, recruited by posting and social groups, this study could be biased in this way. Also, you talk about population of UK junior doctors, but I cannot find the total number of junior doctors in the UK, in order to see if your sample is representative. How did you know the minimum number of junior doctors required for a representative sample? You indicate that the sample was not proportionately representative between women and men. It is possible to indicate the total number of junior doctors women and men in the UK at the Introduction section?	Thank you for the helpful feedback regarding the sample, we have now included a

	reference to the number of junior doctors within the NHS currently. It was not possible to access how many female/male junior doctors there are but we have now included the percentage split of female/male doctors within the introduction. Furthermore, we
--	--

	have included the limitations of the convenience and representativeness of the sample within the limitations of the study.
At the actual moment, there are many studies on the fear, anxiety and psychological distress caused by Covid 19 in healthcare professionals. The most common scales have been Golberg's GHQ12, UWES9 for Work Engagement, or the AMICO scale for fear and anxiety to Covid-19. Did you consider to use some of these scales in your study? You could add some investigations which have been based on the same variables, in order to complete your vision of the healthcare professionals during the pandemic: https://linkprotect.cudasvc.com/url?a=https%3a%2f%2fpubmed.ncbi.nlm.nih.gov%2f34930733%2f&c=E,1,J9LFjN6akWvRQGYThNHHNpJ9MEBUHj7oN9tn9u4pbmuNI8cVQMrsyz79LpwVafBxbR1aCCwsfdE3QkwagPTCEPOSrn_txhQqmxG3jKBY1NT01JE7C23VkoCRZws,&typo=1 https://linkprotect.cudasvc.com/url?a=https%3a%2f%2fwww.ncbi.nlm.nih.gov%2fpmc%2farticles%2fPMC8250215%2f&c=E,1,KV_no7Y9lrFk5N6QHM7ipE98XRaBg2O6_FnehRI9kYmqIqoZ1mt8X8HxXN2vDwVoGnv2FY_Ui3ThayoDhMcYYiilDMCtE12cvJf0kcM1q_6z&typo=1 https://linkprotect.cudasvc.com/url?a=https%3a%2f%2fpubmed.ncbi.nlm.nih.gov%2f33804351%2f&c=E,1,vKod6g95W0_Lp2lvUwTLwruu2bnX8N2Z4L4ACxxPooMrHmWM5qE8KAOfebvEDyg1f_baQlgYrYCgMD2a9AhX-3IH6SOwINjZLv3FpHQ-j59mLiOx&typo=1 https://linkprotect.cudasvc.com/url?a=https%3a%2f%2fpubmed.ncbi.nlm.nih.gov%2f33575245%2f&c=E,1,Js5LMOFTQGQHCTEoxhg__P9VxAi-jg8vAnLqjPz1wDmElev1dqC5OtQMsOz5-Wvpil79E58wCPFvbRvNkLjOrxbopt9ePo3Ynwsvx-yz46YRzRCoBZxTvVyhKnE,&typo=1 https://linkprotect.cudasvc.com/url?a=https%3a%2f%2fpubmed.ncbi.nlm.nih.gov%2f33400325%2f&c=E,1,22yJveUyuc5IUCVqQfbgY_nnuXSp-EVW7wVO0nJIREnL6ZVnX5elrFejywlxmA1s-b5UNbED7ojkNF0YUTAYf_o-Y495B3qOB22RgahUs0OFWzPgBdo19Q,,&typo=1 https://linkprotect.cudasvc.com/url?a=https%3a%2f%2fpubmed.ncbi.nlm.nih.gov%2f32699204%2f&c=E,1,Gpv7mi9qVhu7SP1pKqTIZZ4XMcPxUtt_MfsSFBU7-fuk9NlnE0t4aovUmEPHGpJi4kMijseoSzJ_tMBsbWam92f5RDzbl88hVsMbU_oNPu4MHjOqzsAGk7r48w,,&typo=1	Thank you for the interesting feedback regarding the different wellbeing and work engagement scales, however this was not part of our

	scope as we wanted to look at working conditions in relation to suicide and the HSE measures are more closely linked to that. As mentioned this paper is part of a wider mixed methods study in which one of the papers (to be published) explores the relationship between
--	--

	en psych ologic al distre ss, worki ng condit ions and suicid e. We have never theles s highli ghted the role of positi ve wellb eing as well as an avenu e for future resea rch. The literat ure sugge sted were very helpfu l and sever al of these have now been includ ed and
--	--

	referenced within the paper .
--	-------------------------------